# The Complete Digital Workflow in Fixed Prosthodontics Updated: A Systematic Review

**DOI:** 10.3390/healthcare11050679

**Published:** 2023-02-25

**Authors:** Selina A. Bernauer, Nicola U. Zitzmann, Tim Joda

**Affiliations:** 1Department of Reconstructive Dentistry, University Center for Dental Medicine Basel, University of Basel, 4058 Basel, Switzerland; 2Clinic of Reconstructive Dentistry, Center of Dental Medicine, University of Zurich, 8032 Zurich, Switzerland

**Keywords:** systematic review, fixed prosthodontics, tooth-borne, tooth-supported, implant-supported, complete digital workflow

## Abstract

Digital applications have changed therapy in prosthodontics. In 2017, a systematic review reported on complete digital workflows for treatment with tooth-borne or implant-supported fixed dental prostheses (FDPs). Here, we aim to update this work and summarize the recent scientific literature reporting complete digital workflows and to deduce clinical recommendations. A systematic search of PubMed/Embase using PICO criteria was performed. English-language literature consistent with the original review published between 16 September 2016 and 31 October 2022 was considered. Of the 394 titles retrieved by the search, 42 abstracts were identified, and subsequently, 16 studies were included for data extraction. A total of 440 patients with 658 restorations were analyzed. Almost two-thirds of the studies focused on implant therapy. Time efficiency was the most often defined outcome (*n* = 12/75%), followed by precision (*n* = 11/69%) and patient satisfaction (*n* = 5/31%). Though the amount of clinical research on digital workflows has increased within recent years, the absolute number of published trials remains low, particularly for multi-unit restorations. Current clinical evidence supports the use of complete digital workflows in implant therapy with monolithic crowns in posterior sites. Digitally fabricated implant-supported crowns can be considered at least comparable to conventional and hybrid workflows in terms of time efficiency, production costs, precision, and patient satisfaction.

## 1. Background

The global trend towards digitization dominates all fields of dentistry today. Particularly in fixed prosthodontics, as a technique-oriented discipline, computerized dentistry has enabled new clinical protocols and production processes [1]. While the continuous development of computer-aided design and computer-aided manufacturing (CAD/CAM) techniques is the driving force in dental technology, the adoption of intraoral scanners (IOS) has significantly changed clinical procedures in recent years [2]. Together, these technologies now enable complete digital workflows for single-visit treatments for tooth-borne (tooth-supported) and implant-supported monolithic fixed dental prostheses (FDPs) [2].

Complete digital protocols consist of three main work steps: (i) the 3D acquisition of the individual patient situation directly in the mouth with IOS; (ii) digital design with dental software applications (CAD) for rapid prototyping such as milling or 3D printing (CAM) in a fully virtual environment without any physical dental models (plaster casts); and (iii) clinical delivery of the dental restoration [3]. Crucial steps are the generation, transfer, and further processing of the created IOS data (in Standard Tessellation Language [STL] format) [4]. Overall, the digital workflow is associated with mechanically high-quality monolithic restorations and reproducible fabrication in a simplified process with a reduced need for manual human interaction [5].

In the past, dental research has mostly focused on a single step within this three-step process. Typically, the focus was on in vitro analyses in terms of precision and accuracy, comparing either different IOS systems or rapid prototyping methods for fabricating the final restorations. Besides some single case reports, there was a lack of clinical studies in the dental literature, particularly randomized controlled trials (RCTs) investigating the entire digital workflow [6].

It is important to understand the impact of the current digitalization trend on changing well-established protocols in terms of the clinical and technical feasibility of complete digital workflows, the long-term results, and the economic implications [7,8,9]. In 2017, a systematic review was the first to screen the scientific literature for evidence describing the use of complete digital workflows in fixed prosthodontics for treatment with tooth-borne or implant-supported fixed restorations. This review concluded that the level of evidence for complete digital workflows was low: only three publications investigating single-unit restorations were included, and no studies investigating multi-unit restorations were identified at that time [6].

Advances in the application of digital hardware and software in dentistry occur fast. In recent years, numerous new technologies and commercial products have been released, both for IOS systems and in the CAD/CAM domain. A non-specific PubMed search using the term “digital dentistry” yields 2070 publications for the year 2022. When limited to the year 2017 (the time of the original review), only 953 of the techniques are identified, less than half. Based on this massive increase in such a short period of time, it is of interest to know if the proportion of qualitative clinical trials in fixed prosthodontics has also increased in line with this trend. Therefore, the aim of this systematic review was to update the review originally published in 2017, to present current data describing the latest developments in digitally enhanced fixed prosthodontics, and to derive clinical recommendations for routine use.

## 2. Materials and Methods

### 2.1. Search Strategy and Study Selection

This systematic review is an updated version of a review published in 2017 [6]. The search strategy, based on the PICO criteria, as well as the inclusion criteria are consistent with the previous publication and have been adapted for the new timeframe. The PICO question was formulated as follows [6]: “Is a complete digital workflow with intraoral optical scanning (IOS) plus virtual design plus monolithic restoration for patients receiving prosthodontic treatments with (A) tooth-borne or (B) implant-supported fixed restorations comparable to conventional or mixed analog-digital workflows with conventional impression and/or lost-wax technique and/or framework and veneering in terms of feasibility in general or survival/success-analysis including complication assessment with a minimum follow-up of one year or economics or esthetics or patient-centered factors?” [6].

A systematic electronic search was performed using PubMed, Medline, and Embase for English-language publications. Literature consistent with the original review criteria published between 16 September 2016 and 31 October 2022 was considered. In addition, grey literature, such as Google Scholar, was screened. The search syntax was categorized into population, intervention, comparison, and outcome (PICO). Each category was assembled from a combination of Medical Subject Headings (MeSH Terms) as well as free-text words in simple or multiple conjunctions (Table 1).

Finally, a manual search in the dental literature was also conducted. The following journals were considered: Clinical Implant Dentistry & Related Research, Clinical Oral Implants Research, European Journal of Oral Implantology, Implant Dentistry, International Journal of Oral & Maxillofacial Implants, Journal of Clinical Periodontology, Journal of Computerized Dentistry, Journal of Dental Research, Journal of Oral & Maxillofacial Surgery, Journal of Oral Implantology, Journal of Periodontal & Implant Science, Journal of Periodontology, Journal of Prosthodontics, International Journal of Prosthodontics, Prosthetic Dentistry, and Prosthodontic Research. An additional search of the bibliographies of all full-text articles, selected from the electronic search, was performed.

This systematic review was conducted in accordance with the guidelines of PRISMA (Preferred Reporting Items of Systematic Reviews and Meta-Analyses) [10].

### 2.2. Inclusion Criteria

This systematic review focused on RCTs as the highest level of clinical evidence, particularly those describing complete digital workflows in fixed prosthodontics that analyzed at least one of the following parameters: economics in terms of time and/or cost analyses, esthetics, patient-centered outcomes with or without follow-up, as well as survival and success rate analyses including assessments of complications of at least 1 year under function. The following inclusion criteria were defined [6]:Clinical trials, limited to RCTs with at least 10 patients;Treatment concepts with FDPs, either tooth-borne or implant-supported for single- or multi-unit restorations;Processing of a complete digital workflow (without physical models); andReporting of information on the used clinical work steps and technical production.

### 2.3. Selection of Studies

Title and abstract screening were performed by two independent researchers (S.A.B. and T.J.), who considered the defined inclusion criteria. If the provided information was not sufficient, full texts were retrieved and evaluated by both reviewers. Several publications reported on the same patient population; these publications that summarized different outcomes were merged. Selected articles were subjected to further analyses. Throughout this complete process, disagreements were resolved through discussion.

### 2.4. Data Extraction

The following information were extracted from the included publications: author, year of publication, description of the specific study design, number of patients treated and examined, type of fixed restorations (including the number of abutment teeth and/or dental implants), clinical treatment concept, methodological approach for laboratory processing, description of the material properties, as well as defined primary (and secondary) outcomes.

Finally, all included studies were subdivided into four groups based on the type of prosthetic abutments and the number of units: A1. tooth-borne single crowns; A2. tooth-borne multi-unit FDPs; B1. implant-supported single crowns; and B2. implant-supported multi-unit FDPs. The information extracted from the articles was tabulated, and if possible, a meta-analysis was to be conducted.

## 3. Results

### 3.1. Included Studies

The systematic search was completed on October 31, 2022, and results are current as of this date. Of the 394 titles retrieved by the electronic search, 42 potentially relevant abstracts were identified; however, 28 of these were excluded from the final analysis. In addition, two studies were found through manual search, resulting in a total of 16 studies for data extraction (Figure 1).

The reasons for exclusion were (*n* = 28):No data on complete digital workflows (*n* = 5)Not an RCT (*n* = 17)Workflow did not investigate final prosthetic restorations (*n* = 6)

### 3.2. Descriptive Analysis

The 16 identified RCTs reported on a total of 440 patients, with 236 tooth-borne restorations and 422 implant-supported restorations. Only one of the 16 RCTs included follow-up examinations. General data for study design, type of fixed restoration, number of subjects, and defined outcomes are summarized in Table 2. Based on the prosthetic design, included studies were divided into four groups: (A1) six publications for tooth-borne single-unit restorations [11,12,13,14,15,16,17]; (A2) no publication for tooth-borne multi-unit restorations; (B1) eight publications for implant-supported single-unit restorations [18,19,20,21,22,23,24,25]; and (B2) two publications for implant-supported multi-unit restorations [26,27,28,29].

Due to the heterogeneity of the included RCTs with different study designs and outcomes, a direct comparison among the identified publications was not feasible, and consequently, a meta-analysis could not be performed. Thus, the review of the full texts followed a descriptive analysis. Detailed information of each study, categorized in A1-B1-B2, is shown in Table 3, Table 4 and Table 5. Figure 2 displays the risk of bias for the included studies. No additional analyses were performed.

### 3.3. Group A1—Tooth-Borne Single-Unit Restorations (Table 3)

Six studies compared digital to conventional workflows for tooth-borne single crowns, with a total of 236 prosthetic units. Regarding the precision of marginal fit of the FDPs, three studies found no statistically significant differences of the fabricated crowns between workflows [14,16,17]. One RCT documented a trend towards better marginal fit for the conventional workflow [13]. Another RCT found better marginal and internal adaption for crowns fabricated with digital workflows, but the clinical evaluation showed similar marginal adaptation [12]. Occlusal contacts were found to be better for digitally produced crowns, while no differences were found for marginal fit, proximal contact, and crown morphology [11].

Four studies also investigated time efficiency, with two studies reporting no statistically significant differences in total clinical treatment times [13,15], and one study showing a shorter impression time for IOS compared with a conventional workflow [14]. The other RCT investigated a complete digital workflow, different hybrid workflows with a physical cast, and a conventional workflow as a control [15]. Laboratory fabrication time was significantly shorter for the conventional cast compared to all CAD/CAM casts because the digital workflow included delivery of the CAD/CAM cast from the manufacturer to the dental laboratory. Delivery of the crowns was significantly faster for the fully digital workflow, followed by the conventional workflow.

### 3.4. Group B1—Implant-Supported Single-Unit Restorations (Table 4)

This group comprised eight studies, including 302 patients. A total of 342 implant-supported single crowns were examined. The most frequently considered topic was time efficiency (*n* = 6), followed by precision (*n* = 5), patient satisfaction (*n* = 4), esthetics (*n* = 4), marginal-bone loss (*n* = 2), and cost efficiency (*n* = 1). For economic analyses, all studies that examined time efficiency found significantly higher time savings for digital workflows (*n* = 6/6 studies), and costs were also significantly lower for the complete digital approach (*n* = 1/1 study). Patient satisfaction was rated significantly better for digital solutions in most publications (*n* = 3/4 studies). For the three other parameters (precision, esthetic outcome, and marginal bone loss), no significant differences between the workflows were reported.

### 3.5. Group B2—Implant-Supported Multi-Unit Restorations (Table 5)

Two studies investigated implant-supported multi-unit restorations with a total of 30 patients. Eighty implant-supported three-unit FDPs were fabricated either with digital or conventional workflows. Both studies examined time efficiency. Hashemi et al. stated significant less mean laboratory time and a shorter total fabrication time for the digital workflow [28]. Joda et al. [29] investigated time efficiency of two different digital and one conventional workflow. Significant differences in time efficiency for pairwise comparisons of the total work time were observed. The proprietary digital workflow *3Shape* (IOS: TRIOS 3 Pod) was shown to be more time-efficient than the conventional workflow, while the proprietary digital workflow *Dental Wings* (IOS: Virtuo Vivo) required more time. The cost analysis was favored the digital workflow, with significantly lower production costs for completely digital fabricated FDPs [29]. Based on the same study population, patient-centered outcomes and clinical performance were also investigated [27]. Patient satisfaction with the final monolithic ZrO2 FDPs, as assessed in a double-blind testing, revealed no significant difference between the different workflows, but significantly lower overall ratings were reported by the dental professional than by patients [27]. Finally, the mean total chairside adjustment time, as the sum of interproximal, pontic, and occlusal corrections, was not significantly different among all three workflows [26].

## 4. Discussion

This systematic review aimed to summarize recent data from RCTs on conventional versus complete digital workflows for fabrication of FDPs. Overall, data from 658 FDPs from 16 RCTs were summarized. The review found ambiguous results for clinical parameters in tooth-borne single-unit restorations, while for implant-supported single- and multi-unit restorations, significantly shorter fabrication time at lower costs was demonstrated for digital compared to conventional workflows.

The systematic search strategy and inclusion criteria used in the present review were identical to those used in the previous review [6], only the time frame was adjusted. In general, RCTs provide the best clinical evidence for generating a systematic review, and only this study type was included. Consequently, the number of included publications was smaller than if all study types had been considered. Compared with the previous review that covered publications up to September 2016 and identified three RCTs, the present systematic review covering the last 6 years identified five times as many RCTs, including two studies of multi-unit restorations. Nevertheless, this still represents relatively few studies compared with the hundreds of publications that report FDP treatment using purely conventional protocols. This suggests that the long-predicted hype for digitization in the MedTech industry has yet to be realized.

Interestingly, the number of studies in the subgroups A1-B1-B2 showed a heterogeneous distribution, and no RCT was identified that investigated a complete digital workflow for tooth-borne multi-unit restorations. Most included RCTs (10/16; 63%) investigated implant-supported restorations. Possible explanations for this include a general trend towards more implant-driven treatment concepts, the fact that complete digital workflows are simply predestined for monolithic restorations on implants, or that the funding of clinical trials by industrial sponsors favors implant concepts.

Most of the identified RCTs (12/16; 75%) focused on time efficiency as an economic key performance indicator [11,13,14,15,18,21,22,23,24,25,28,29]. Eleven studies (69%) investigated the precision of the FDPs [11,12,13,14,16,17,19,22,23,24,25] and five studies (32%) analyzed patient satisfaction [16,18,20,21,27]. This shows that patient-centered parameters are becoming increasingly important, in addition to the classic clinical parameters in fixed prosthodontics, such as analyses of marginal integrity and occlusion in the overall context of precision [30].

From an economic point of view, complete digital workflows could demonstrate a clear advantage over conventional procedures. This was regardless of whether the restorations were tooth-borne or implant-supported, and regardless of the size of the restorations, as single crowns or multi-units [11,13,14,15,18,21,22,23,24,25,28,29]. In terms of precision, both workflows seemed to offer similar performance, with a possible slight advantage for well-established conventional protocols over digital workflows for treatment with tooth-borne restorations [11,12,13,14,16,17,19,22,23,24,25]. Finally, the patients either did not notice any differences between digitally or conventionally produced FDPs or they rated the restorations from the digital workflows better [16,18,20,21,27].

The correct application of a workflow (digital or conventional) to an appropriate indication is crucial for the success of the overall prosthetic therapy and for a satisfied patient [31]. For digital processing, a teamwork approach is particularly important—this equally includes the clinician, the dental assistants, and the technician [32]. The complete digital workflow has the potential to become a game-changer in (fixed) prosthodontics [33].

Nevertheless, the conventional workflow remains the current gold standard. In recent years, individual components in the workflow have increasingly become digitized for both tooth-borne and implant-supported restorations. This digital change began in dental technology with the introduction of CAD/CAM technology. As a consequence, the technical-dental protocol has been transformed to a hybrid analog-digital workflow. For the indication of single crowns, especially on implants in the posterior region, there seems to be a strong trend in favor of complete digital workflows with monolithic restorations and pre-fabricated titanium base abutments [34]. Subsequently, IOS ideally has completed the clinical gap [35]. The continuous development of digital scanning techniques has enabled quick, safe, and patient-friendly 3D capturing of the clinical situation [36,37]. Use of IOS is particularly beneficial in implant therapy because it is not necessary to optically record an individual preparation margin on the tooth, but only a standardized supra-mucosal localized scanbody. For single-unit restorations, the digital bite registration is much easier and more reproducible than for multi-units [38]. Finally, economic factors offering high-level quality restoration with reduced treatment time and lower production costs are the biggest driver [39,40].

## 5. Conclusions

Based on the findings of this systematic review, it can be concluded that the amount of qualitative clinical research investigating complete digital workflows has increased within the last 6 years. However, the absolute number of RCTs, in particular those investigating treatment with multi-unit restorations, is still low. Good quality clinical evidence exists supporting the use of complete digital workflows in implant therapy with monolithic crowns in posterior sites. Digitally fabricated implant-supported single units can be considered at least comparable to conventional and hybrid workflows in terms of time efficiency, production costs, precision, and patient satisfaction.

Future clinical research based on RCTs is imperative to gain clarity on the clinical performance of digital workflows. The difficulty is the rapid digital evolution, so that the devices and tools from the clinical trials are already “obsolete” after 1 to 2 years (when the data are published). It is therefore particularly important that the versions of hardware and software used are always specified.

## Figures and Tables

**Figure 1 healthcare-11-00679-f001:**
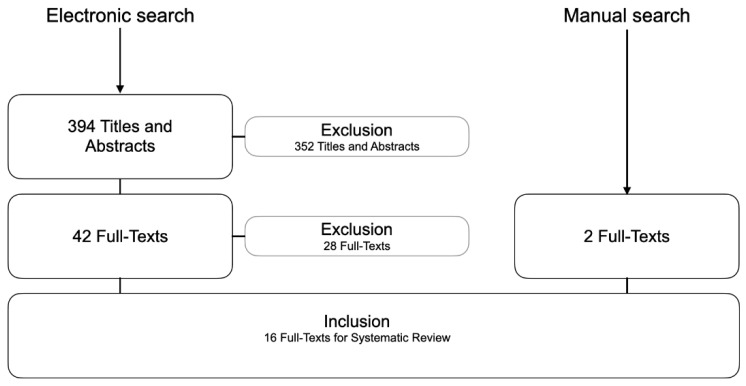
Flow-chart showing the electronic and manual search results.

**Figure 2 healthcare-11-00679-f002:**
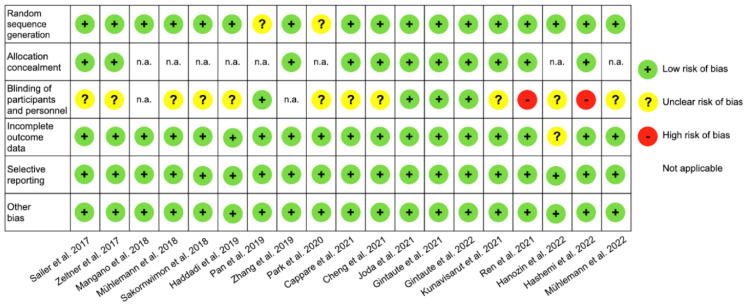
Presentation of risk of bias assessments for included studies according to the Cochrane collaboration’s tool.

**Table 1 healthcare-11-00679-t001:** Overview of the electronic search strategy according to PICO criteria.

**Focused question (PICO)**		Is a complete digital workflow with intraoral optical scanning (IOS) plus virtual design plus monolithic restoration for patients receiving prosthodontic treatments with (A) tooth-borne or (B) implant-supported fixed restorations comparable to conventional or mixed analog-digital workflows with conventional impression and/or lost-wax-technique and/or framework and veneering in terms of feasibility in general or survival/success-analysis including complication assessment with a minimum follow-up of one year or economics or esthetics or patient-centered factors?
**Timeline**		From 16 September 2016 until 31 October 2022
**Search Strategy**	**Problem**	{(“Dental Prosthesis” [6]) OR (“Crowns” [6]) OR (“Dental Prosthesis, Implant-Supported” [6] OR (“Crowns, Implant-Supported” [6]) OR (crown) OR (fixed dental prosthesis) OR (fixed reconstruction) OR (fixed restoration) OR (dental bridge) OR (implant crown) OR (implant prosthesis) OR (implant restoration) OR (implant reconstruction)}
	**Intervention**	{(“Computer-Aided Design” [6]) OR (digital workflow) OR (digital technology) OR (computerized dentistry) OR (intraoral scan) OR (digital impression) OR (scanbody) OR (virtual design) OR (digital design) OR (cad/cam) OR (rapid prototyping) OR (monolithic) OR (full-contour)}
	**Control**	{(“Dental Technology” [6]) OR (conventional workflow) OR (lost-wax-technique) OR (porcelain-fused-to-metal) OR (PFM) OR (implant impression) OR (hand-layering) OR (veneering) OR (framework)}
	**Outcome**	{(“Study, Feasibility” [6]) OR (“Survival” [MeSH]) OR (“Success” [MeSH]) OR (“Economics” [MeSH]) OR (“Costs, Cost Analysis” [MeSH]) OR (“Esthetics, Dental” [MeSH]) OR (“Patient Satisfaction” [MeSH]) OR (feasibility) OR (efficiency) OR (esthetics) OR (patient-centered outcome)}

**Table 2 healthcare-11-00679-t002:** General data of the included trials, including study design, type of fixed restoration, number of investigated subjects, and defined outcome(s).

No.	Study (Year)	Author	Study Design	Type of Restoration	Number of Subjects	Outcome
1.	2016 2016	Seiler et al. [15] Zeltner et al. [17]	RCT − Five-armed design − Examiner-blinded	A1. Tooth-borne crowns − *n* = 10: LS2 crowns fully digital workflow. − *n* = 30: LS2 crowns digital workflow with physical cast. − *n* = 10: LS2 crowns conventional workflow.	10 patients 50 crowns	− Time efficiency [15] − Precision [17]
2.	2018	Mangano et al. [21]	RCT − Two-armed design	B1. Implant-supported crowns − *n* = 25: ZrO2 crowns digital workflow (CS 3600 Carestream Dental). − *n* = 25: Metal-ceramic crowns conventional workflow (Control: Elite HD Plus, Zhermack).	50 patients 50 crowns	− Primary outcome: Implant crown success and complications, peri-implant marginal bone loss − Secondary outcome: Patient satisfaction and time and cost efficiency
3.	2018	Mühlemann et al. [13]	RCT − Five-armed design − Blinded	A1. Tooth-borne crowns − *n* = 40: LS2 crowns digital workflow. − *n* = 10: LS2 crowns conventional workflow.	10 patients 50 crowns	− Primary outcome: Precision − Secondary outcome: Time efficiency
4.	2018	Sakornwimon et al. [16]	RCT − Two-armed design − Examiner-blinded	A1: Tooth-borne crowns − *n* = 16: ZrO2 crowns digital workflow. − *n* = 16: ZrO2 crowns conventional workflow.	16 patients 32 crowns	− Precision − Self-perception of time involved, taste/smell, occlusal registration, size of impression tray/scanner, gag reflex, and overall preference
5.	2019	Haddadi et al. [12]	RCT − Two-armed design	A1. Tooth-borne crowns − *n* = 19: LS2 crowns digital workflow (Trios 3, 3shape.) − *n* = 19: LS2 crowns conventional workflow (Extrude, Kerr, Orange, USA).	19 patients 38 crowns	− Precision
6.	2019	Pan et al. [23]	RCT − Two-armed design − Double-blinded	B1. Implant-supported crowns − *n* = 40: ZrO2 crowns digital workflow (Trios, 3Shape). − *n* = 40: ZrO2 crowns conventional workflow (Impregum Penta, 3m ESPE GmbH).	40 patients 80 crowns	− Precision − Time efficiency
7.	2019	Zhang et al. [25]	RCT − Two-armed design	B1. Implant-supported crowns − *n* = 16: Monolithic LS2 crowns digital workflow. − *n* = 17: ZrO2 framework and ceramic veneering.	33 patients 33 crowns	− Precision − Time efficiency
8.	2020	Park et al. [14]	RCT − Three-armed design − Cross-over design	A1. Tooth-borne crowns − *n* = 13: LS2 crowns digital workflow. − *n* = 13: LS2 crowns conventional workflow.	13 patients 26 crowns	− Primary outcome: Precision, time efficiency − Secondary outcome: Accuracy of IOS
9.	2021	Cappare et al. [18]	RCT − Two-armed design − Blinded − 12 month follow up	B1. Implant-supported crowns − *n* = 25: Provisional crowns delivered directly after implant placement + ZrO2 crowns digital workflow. − *n* = 25: Provisional crowns delivered directly after implant placement + ZrO2 crowns conventional workflow.	50 patients 50 crowns	− Primary outcome: Marginal bone loss, pink esthetic score − Secondary outcome: Time efficiency, Patient satisfaction
10.	2021	Cheng et al. [11]	RCT − Four-armed-design	A1. Tooth-borne crowns − *n* = 10: PMMA interim crowns digital workflow made by experienced clinicians. − *n* = 10: PMMA interim crowns digital workflow made by less experienced clinicians. − *n* = 10: PMMA interim crowns conventional workflow made by experienced clinicians. − *n* = 10: PMMA interim crowns conventional workflow made by less experienced clinicians.	40 patients 40 (interim) crowns	− Precision − Time efficiency
11.	2021 2021 2022	Joda et al. [29] Gintaute et al. [26] Gintaute et al. [27]	RCT − Three-armed-design − Double-blinded	B2. Implant-supported three-unit restorations. − *n* = 20: ZrO2 FDPs digital workflow (Test 1: Trios 3, 3Shape). − *n* = 20: ZrO2 FDPs digital workflow (Test 2: Virtuo Vivo, Dental Wings). − *n* = 20: ZrO2 FDPs hybrid workflow with conventional impression and digitized casts (Control).	20 patients 60 three-unit FDPs	− Primary outcome: Time efficiency [29] − Secondary outcome: Cost efficiency − Primary outcome: Patient satisfaction [27] − Primary outcome: Clinical performance [26]
12.	2021	Kunavisarut et al. [20]	RCT − Two-armed design − Examiner-blinded	B1. Implant-supported crowns − *n* = 10: LS2 crowns digital workflow. − *n* = 10: Polymer-infiltrated ceramic network crowns digital workflow. − *n* = 10: LS2 crowns conventional workflow. − *n* = 10: Polymer-infiltrated ceramic network crowns conventional workflow.	40 patients 40 crowns	− Patient satisfaction
13.	2021	Ren et al. [24]	RCT − Two-armed design	B1. Implant-supported crowns − *n* = 20: Provisional + ZrO2 crowns IOS and digital workflow. − *n* = 20: Provisional + ZrO2 crowns with conventional impressions and hybrid workflow.	40 patients 40 crowns	− Primary outcome: Precision − Secondary outcome: Time efficiency
14.	2022	Hanozin et al. [19]	RCT − Two-armed design	B1. Implant-supported crowns − *n* = 9: CAD/CAM provisional crowns prepared prior to surgery and immediate restoration. − *n* = 9: Stratified provisional crowns based on a conventional impression 10 days after surgery.	18 patients 18 crowns	− Precision
15.	2022	Hashemi et al. [28]	RCT − Two-armed-design	B2. Implant-supported three-unit restorations − *n* = 10: Digital workflow. − *n* = 10: Hybrid workflow.	10 patients 20 three-unit FDPs	− Primary outcome: Accuracy of IOS − Secondary outcome: Time efficiency
16.	2022	Mühlemann et al. [22]	RCT − Two-armed design	B1. Implant-supported crowns − *n* = 12: Individual Ti abutment + ZrO2 crowns digital workflow centralized CAM. − *n* = 19: Standard titanium abutment + ZrO2 crowns hybrid workflow laboratory CAM.	31 patients 31 crowns	− Precision − Time efficiency

RCT, randomized controlled trial; CAD/CAM, computer-aided design/computer-aided manufacturing; FDP, fixed dental prostheses; IOS, intraoral scanning; LS2, lithium disilicate; ZrO2, zirconium dioxide; PMMA, polymethyl methacrylate acrylic.

**Table 3 healthcare-11-00679-t003:** Detailed study information according to the type of restoration A1 (tooth-borne single-units).

No.	Study	Number of Subjects	Number of Prosthetic Units	Number of Abutment Teeth	Workflow and Materials	Results
1.	2017, Haddadi et al. [12]	*n* = 19	*n* = 38 [19 + 19]	*n* = 38	Digital: Tooth-borne premolar or molar crowns; digital impressions (Trios 3, 3shape); design (Dental System design software, 3Shape, Denmark); milling of LS2 crowns (Röders RXD5, Röders GmbH, Soltau, Germany); evaluation. Conventional: Tooth-borne premolar or molar crowns; conventional impressions (Extrude, Kerr, Orange, USA); fabrication of stone casts; labside scanning (D640, 3Shape, Denmark); design (Dental System design software, 3Shape, Denmark); milling of LS2 crowns (Röders RXD5, Röders GmbH, Soltau, Germany); evaluation.	*Precision:* − Median gap of crowns 60 μm for digital and 78 μm for conventional workflow. Significant better accuracy of IOS at all points except at the cusp tip. − 6 and 12 months follow up clinical re-evaluations; no statistically significant difference between the two impression methods.
2.	2017, Sailer et al. [15] 2018, Zeltner et al. [17]	*n* = 10	*n* = 50 [10 + 10 + 10 + 10 + 10]	*n* = 10	Digital: Group CiL: Tooth-borne premolar or molar crowns; digital impressions (Cerec Bluecam, Dentsply Sirona); CAD software (Cerec Connect software and Cerec inLab 3D, Dentsply Sirona); Labside milling of LS2 crowns (Cerec inLab MC XL milling unit, Dentsply Sirona); evaluation. Digital with physical cast: Tooth-borne premolar or molar crowns. − Group L: Digital impression (Lava C.O.S. 3M ESPE); shipping of the physical cast; CAD software (Lava C.O.S. Software, 3M ESPE, and Cares Software Cares Visual 6.2, Straumann AG); centralized milling process (Straumann AG). − Group iT: Digital impression (Cadent iTero, Align Technologies Inc); shipping of the physical cast; CAD software (Cares Visual 6.2; Straumann AG); centralized milling process (Straumann AG). − Group CiD: Digital impression (Cerec Bluecam, Dentsply Sirona); shipping of the physical cast; CAD software (Cerec Connect software and Cerec inLab 3D, Dentsply Sirona); centralized milling process (infiniDent; Dentsply Sirona). Conventional: Group K: Tooth-borne premolar or molar crowns; conventionally acquired impressions; fabrication of dental stone cast (Quadro-rock Plus; Picodent); waxing; investing; heat pressing the lithium disilicate glass ceramic blank (IPS e.max Press; Ivoclar Vivadent AG); devesting.	*Precision:* − No significant differences between conventional and digital workflows in terms of marginal fit (*p* > 0.05). − In occlusal regions, conventionally manufactured crowns revealed better fit (*p* > 0.05). − Chairside milling resulted in less favorable crown fit than centralized milling production. *Time efficiency:* − Fabrication time for conventional cast was significantly shorter as compared with all CAD/CAM casts. − Conventional crown design (waxing) required significantly more time than all virtual designs (*p* < 0.001). − Delivery of crowns was faster in group CiL (fully digital workflow), followed by the conventional group K (*p* < 0.001). − No statistically significant differences in mean treatment times for the chairside adjustments or total treatment times during the first clinical evaluation (*p* > 0.05). − Significantly less time to finalize the conventionally fabricated crowns compared to most CAD/CAM crowns (group K vs. groups L, CiL, and CiD; *p* < 0.05); finalization of the crowns took significantly more time in group CiD than in any other group (*p* < 0.001).
3.	2017, Sakorniwomo et al. [16]	*n* = 16	*n* = 32 (16 + 16)	*n* = 16	Digital: Tooth-borne molar crowns; digital impressions; design (3shape); milling of monolithic ZrO2 crowns (Lava Plus High Translucency Zirconia, 3M ESPE; hiCut CNC, Hint-Els); clinical evaluation. Conventional: Tooth-borne molar crowns; conventional impressions (Express xT Putty Soft and Express XT Light Body, 3M ESPE); fabrication of stone casts; labside scanning (D900L Scanner, 3Shape); design (3Shape); milling of monolithic ZrO2 crowns (Lava Plus High Translucency Zirconia, 3M ESPE; hiCut CNC, Hint-Els); clinical evaluation.	*Precision:* − No significant differences in clinical marginal fit of ZrO2 crowns fabricated from either digital or conventional impressions (*p* > 0.05). *Patient satisfaction:* − 15 of the 16 patients preferred IOS compared with conventional impressions (*p* < 0.05).
4.	2018, Mühlemann et al. [13]	*n* = 10	*n* = 50 [10 + 10 + 10 + 10 + 10]	*n* = 10	Digital: Tooth-borne crowns; complete digital workflow with four different methods: − Impression-taking and manufacturing process via Lava C.O.S. and CARES CAD software, centralized CAM. − Impression-taking and manufacturing process via Cadent iTero, CARES CAD software, centralized CAM. − Impression-taking and manufacturing process via Cerec Bluecam, Cerec Connect CAD software, laboratory-based CAM. − Impression-taking and manufacturing process via Cerec Bluecam, Cerec Connect CAD software, centralized CAM. Conventional: Tooth-borne crowns; conventional impressions (President, Coltene); stone casts; production of ceramic crown using lost-wax technique.	*Precision:* − No statistically significant differences between groups at any state (*p* < 0.05); trend toward better marginal adaption for conventionally fabricated crowns. *Time efficiency:* − Total clinical treatment time did not show statistical differences (*p* > 0.05).
5.	2020, Park et al. [14]	*n* = 13	*n* = 26 [13 + 13]	*n* = 13	Digital: Tooth-borne crowns; complete digital workflow (AEGIS.PO, Digital Dentistry Solution, CEREC Omnicam, Sirona); design (DESIGN + Suite, Digital Dentistry Solution) and milling (SPEED +, Digital Dentistry Solution) of LS2 crowns (IPS e.max CAD; Ivoclar Vivadent). Conventional: Tooth-borne crowns; conventional impressions; fabrication and scan of master casts (Identica Hybrid, Medit); design (DESIGN + Suite, Digital Dentistry Solution) and milling (SPEED +, Digital Dentistry Solution) of LS2 crowns (IPS e.max CAD; Ivoclar Vivadent).	*Precision:* − No statistically significant differences regarding fit of restorations and accuracy of IOS (*p* < 0.05). *Time efficiency:* − IOS required significantly shorter impression times (AEGIS 7.16 ± 1.50 min and CEREC 7.29 ± 2.03 min) compared to conventional impression taking (12.41 ± 1.16 min).
6.	2021, Cheng et al. [11]	*n* = 40	*n* = 40 [20 + 20]	*n* = 40	Digital: Tooth-borne interim crowns; impression taking (CS 3500, Carestream Dental); CAD (Exocad, Exocad GmbH); milling out of PMMA (PMMA Disk, Ymahachi Dental); clinical evaluation. Conventional: Tooth-borne interim crowns; conventional impression (Cavex CA37, Cavex); diagnostic wax-up and fabrication of vacuum formed translucent-matrices; direct interim crowns on the abutment tooth using PMMA (ALIKE, GC) and the vacuum-formed matrix; clinical evaluation.	*Precision:* − Occlusal contacts better for digitally fabricated interim crowns (*p* = 0.005), no differences for marginal fit, proximal contact, crown morphology (*p* > 0.5). *Time efficiency:* − Mean laboratory and clinical time was significantly less for digital workflow (64.9 ± 16.0 min vs. 128.9 ± 37.0 min). − Significant difference between experienced and less-experienced clinicians in terms of clinical time with the conventional workflow (34.59 ± 8.6 min vs. 50.7 ± 18.1 min), but not in terms of laboratory time (85.0 ± 24.2 min vs. 87.6 ± 39.0 min). − For less-experienced clinicians, overall work steps time was reduced by using the digital workflow (*p* < 0.001).

CAD/CAM, computer-aided design/computer-aided manufacturing; COS, Chairside Oral Scanner; FDP, fixed dental prostheses; IOS, intraoral scanning; LS2, lithium disilicate; ZrO2, zirconium dioxide; PMMA, polymethyl methacrylate acrylic; vs., versus.

**Table 4 healthcare-11-00679-t004:** Detailed study information according to the type of restoration B1 (implant supported single-units).

No.	Study	Number of Subjects	Number of Prosthetic Units	Number of Implant Abutments	Workflow and Materials	Results
1.	2018, Mangano et al. [21]	*n* = 50	*n* = 50 [25 + 25]	*n* = 50	Digital: Impression-taking (CS3600, Carestream Dental, Rochester, NY, USA); design of individualized ZrO2 abutments and temporary PMMA crowns (Exocad Dental CAD); replacement of interim crowns after 2 months with monolithic ZrO2 crowns (Katana, Kuraray Noritake). Conventional: Impression-taking (Elite HDPlus, Zhermack); plaster models; dental technician prepared the Ti-abutment, temporary crowns and wax-up for the metal structures; replacement of interims after 2 months following second impressions with polyvinyl siloxane over metal copings; veneering of the metal structures; application of the final metal-ceramic crowns.	*Cost efficiency:* − Digital procedure presented lower costs than the analog (€ 277 vs. €392). *Peri-implant marginal bone loss:* − No statistically significant differences regarding peri-implant marginal bone loss (average difference of -0.16 mm in favor of the test group; *p* = 0.008). *Patient satisfaction:* − Overall comfort during the impression procedure was better in the digital workflow (*p* < 0.001). *Time efficiency:* − Active working time for the dental technician in digital workflow was more time-efficient than conventional, for provisional (70 ± 15 min vs. 340 ± 37 min; *p* < 0.0001) and final crowns (29 ± 9 min vs. 260 ± 26 min; *p* < 0.0001).
2.	2019, Pan et al. [23]	*n* = 40	*n* = 80 [40 + 40]	*n* = 80	Digital: Impression-taking immediately after implant placement (Trios, 3Shape); fabrication of screw-retained monolithic ZrO2 (Zenotec select hybrid, Wieland Dental); milling and sintering (Zenotec select hybrid, Wieland Dental). Conventional: Conventional impressions 3 months after implant placement (Impregum Penta, 3M ESPE); fabrication and digitization of stone models with lab-scanner (D3000, 3Shape), milling and sintering of screw-retained monolithic ZrO2 crowns (Zenotec select hybrid, Wieland Dental); adjustments by dental technician.	*Precision:* − Clinical evaluation resulted in similar quality of outcomes regarding interproximal and occlusal contact. *Time efficiency:* − Significant differences for total mean clinical chairside time for the digital vs. the conventional workflow (23.2 min vs. 25.7 min; *p* = 0.013). − Digital impression took significantly less time than the conventional method (10.9 min vs. 14.3 min; *p* < 0.001). − Model-free digital workflow took significant less laboratory time (13.6 min vs. 29.9 min; *p* < 0.05). − No significant difference in the mean clinical chairside adjustment time at crown delivery (12.3 min vs. 11.4 min).
3.	2019, Zhang et al. [25]	*n* = 33	*n* = 33 [17 + 16]	*n* = 33	Digital: Digital impressions; design of the crown (CEREC Omnicam, Sirona, Dentsply); milling (CEREC MC XL Premium, Sirona, Dentsply) of monolithic LS2-crowns (IPS e.max CAD, Ivoclar Vivadent). Conventional: Silicone impressions (Silagum, DMG); fabrication and scan of stone models; milling and sintering of ZrO2 frameworks and ceramic veneering.	*Precision:* − Test group demonstrated fewer adjustments and showed better fabricating accuracy compared with the control group (median adjustment count was 2.00 ± 1.09 in test and 3.00 ± 1.05 in control; *p* = 0.001). *Time efficiency:* − The total active working time/total time for two workflows was 92.3/113.7 min for the test group and 146.3/676.3 min for the control group. − Complete digital workflows had significant shorter clinical and laboratory times (40.2 ± 8.7 vs. 89.9 ± 12.2; *p* < 0.0001).
4.	2021, Cappare et al. [18]	*n* = 50	*n* = 50 [25 + 25]	*n* = 50	Digital: Implant and temporary abutment insertion; impression recorded using CAD/CAM system (Cerec Omnicam, Dentsply Sirona); fabrication of temporary prosthesis in PMMA (Sirona Cerec MCXL milling machine, Dentsply Sirona); four months after final digital impressions were recorded (Cerec Omnicam, Dentsply Sirona); insertion of final prosthesis in zirconia ceramic. Conventional: Implant and temporary abutment insertion; pre-fabricated acrylic resin crowns were obtained and adapted with an auto-polymerizing acrylic resin (Duralay, Reliance Dental Manufacturing LLC) along the margins of the temporary abutment; after 4 months, final impressions were taken using polyether (Impregum Penta, 3M ESPE); insertion of final ZrO2 crowns.	*Esthetics:* − No significant differences in the mean total pink esthetic score; test (7.75 ± 0.89) and control (7.5 ± 0.81). *Patient satisfaction:* − Patients preferred digital workflow over the conventional workflow (97.6 ± 4.3 vs. 69.2 ± 13.8, *p* = 0.005). *Peri-implant bone loss:* − Mean bone loss at 12 month follow-up of 0.12 ± 0.66 for digital workflow vs. 0.15 ± 0.54 mm for conventional workflow; no statistically significant differences between the two workflows. *Time efficiency:* − Patients felt that the time spent on the workflow was justified; digital workflow was more time-efficient than the conventional workflow (97.2 ± 7.3 vs. 81.2 ± 11.3; *p* = 0.023).
5.	2021, Kunavisarut et al. [20]	*n* = 40	*n* = 40 [10 + 10 + 10 + 10]	*n* = 40	Digital: Digital impressions (Trios, 3Shape); division into subgroups according to the restorative material: LS2 (N!CE, Straumann) or PICN (Enamic, Vita); chairside design and production (CARES Visual Chairside; C-Series CAD/CAM Milling, Straumann); bonding to Ti-base (Variobase, Straumann); clinical try-in and adjustments. Conventional: Conventional closed tray silicone impressions (Impregum, 3M Espe); digitalization of master casts lab-scanner (D900L, 3Shape); division into subgroups according to the restorative material: LS2 (N!CE, Straumann) or PICN (Enamic, Vita); chairside design and production (CARES Visual Chairside; C-Series CAD/CAM Milling, Straumann); bonding to Ti-bases (Variobase, Straumann); clinical try-in and adjustments.	*Patient satisfaction:* − Impression techniques: IOS demonstrated significantly less taste irritation than conventional impressions (*p* = 0.036) − Homogenous results for both impression procedures in terms of duration of procedures, comfort, level of anxiety, nausea, and pain − No significant differences for comparisons between LS2 and PICN crowns
6.	2021, Ren et al. [24]	*n* = 40	*n* = 40 [20 + 20]	*n* = 40	Digital: Digital impression (Trios, 3Shape); Ti-abutment and ZrO2 crowns were designed and milled (Organical Multi 5X, Organical CAD/CAM GmbH); a dental technician polished and refined the milled abutments and crowns; IOS before and after clinical adjustment of the crowns (Trios, 3Shape); STL files were analyzed with Geomagic or crown adjustment evaluation. Conventional: Conventional silicone impressions; digitalization of master casts by lab-scanner [3Shape]; Ti-abutments were designed and milled (Organical Multi 5X, Organical CAD/CAM GmbH); a dental technician adjusted the abutments; new model scan, crowns were designed and milled (Organical Multi 5X, Organical CAD/CAM GmbH); refined by a dental technician; IOS before and after clinical adjustments of the crowns (Trios, 3Shape); STL files were analyzed with Geomatic or crown adjustment evaluation.	*Precision:* − Significant differences in crown adjustments; the complete digital workflow had better precision, particularly on the occlusal surface [−212.7 ± 150.5 and −330.7 ± 192.5 μm in the test and control groups, respectively (*p* = 0.037)] *Time efficiency:* − The mean chair-side time was 20.20 ± 3.00 and 26.65 ± 4.53 min in the test and control groups, respectively (*p* < 0.001) − The mean laboratory time was 43.70 ± 5.56 and 84.55 ± 5.81 min in the test and control groups, respectively (*p* < 0.001) − Complete digital workflows had significant shorter clinical and laboratory times.
7.	2022, Hanozin et al. [19]	*n* = 18	*n* = 18 [9 + 9]	*n* = 18	Digital: Digital impression (Trios, 3Shape); Digital wax-up (CARES software) for implant planification (coDiagnostiX), digitally design of custom-made ZrO2 abutment (CARES X-Stream abutment, Straumann) and CAD/CAM PMMA crown; fully guided implantation; digital impression with scanbody (Trios, 3Shape); clinical check of the final ZrO2 abutment and provisional crown with immediate loading. Conventional: alginate impressions; digital implant planning (coDiagnostiX) based on a conventional wax-up; free-hand surgical implantation, conventional impressions with open tray, design of the final ZrO2 abutment and PMMA crown; insertion 10 days postoperative.	*Esthetic:* − White esthetic score was comparable in both groups (*p* = 0.45), trend to higher score for the conventional workflow. − Tendency for higher pink esthetic score for the test group (*p* = 0.057). *Precision:* − No significant differences regarding precision needed (occlusion: *p* = 0.70, interproximal contact: *p* = 0.69), but in about half of the cases in both groups adjustments were necessary. *Patient satisfaction:* − IOS impressions were significantly more comfortable compared to conventional impressions (*p* = 0.014).
8.	2022, Mühlemann et al. [22]	*n* = 31	*n* = 31 [12 + 19]	*n* = 31	Digital: Digital impressions (Trios 3, 3Shape); scan data were uploaded to a centralized server (Virtual Atlantis Design, Dentsply Sirona); remote validation by dental technician and centralized CAM of the abutment (Atlantis, CustomBase solution, Dentsply Sirona) and ZrO2 crowns (Atlantis Crown, Full-contour, Dentsply Sirona); ZrO2 crowns in sintered stage, customized Ti-abutment and digital models were shipped; crowns were prepared for try-in by temporarily cementing on abutments; clinical evaluation; finalization by dental technician. Hybrid: Conventional impressions (Permadyne, 3M, ESEP GmbH); fabrication of stone models, digitalization with lab-scanner (Ceramill Map 400, Amann Girrbach); in house CAD (Ceramill, Amann Girrbach) of monolithic crowns on Ti-base abutments; crowns were prepared for try-in by temporarily cementing on abutments; clinical evaluation; finalization by dental technician.	*Precision:* − At try-in and delivery, efficacy of prosthetic manufacturing was similarly high in both workflows. *Time efficiency:* − Mean total impression time was shorter for digital impressions (9.5 ± 3.5 min) compared to conventional impressions (15.1± 4.6 min) (*p* < 0.0001). − Mean total working time of the dental technician for centralized complete digital workflow (131 ± 31 min) and hybrid workflow (218 ± 31 min) (*p* < 0.0001). − Mean total waiting time for centralized complete digital workflow (8593 ± 4407 min) compared to the hybrid workflow (764 ± 65 min) (*p* < 0.0001)

CAD/CAM, computer-aided design/computer-aided manufacturing; FDP, fixed dental prostheses; IOS, intraoral scanning; LS2, lithium disilicate; PICN, polymer infiltrated ceramic network; PMMA, polymethyl methacrylate acrylic; STL, stereolithography files; vs., versus; ZrO_2_, zirconium dioxide.

**Table 5 healthcare-11-00679-t005:** Detailed study information according to the type of restoration B2 (implant supported multi units).

No.	Study	Number of Subjects	Number of Prosthetic Units	Number of Implant Abutments	Workflow and Materials	Results
1.	2021, Joda et al. [29] 2021, Gintaute et al. [26] 2022, Gintaute et al. [27]	*n* = 20	*n* = 60 [20 + 20 + 20]	*n* = 40	Digital: Digital impressions; model-free fabrication of three-unit monolithic ZrO2 iFDPs using two different IOS systems; Test 1: (Trios 3, 3Shape), and Test 2 (Virtuo Vivo, Dental Wings) including company-related CAD/CAM lab software; milling of the three-unit monolithic ZrO2 iFDPs (Ceramill 2 Motion, Amann Girrbach), clinical assessment of restorations. Conventional: Classical impression-taking (Impregum, 3M ESPE), digitization of the gypsum casts with lab-scan (Ceramill Map 400+, Amann Girrbach), Exocad Lab-Software, milling of three-unit monolithic ZrO2 iFDPs (Ceramill 2 Motion, Amann Girrbach), clinical assessment of restorations.	*Time efficiency:* − Significant differences in time efficiency for pairwise comparison of the total work time. Test 1 demonstrated the best performance for time-efficiency (97.5 min). *Cost efficiency:* − Digital workflow resulted in significantly lower production costs compared to the mixed analog-digital workflows (Test 1: 566 CHF; Test 2: 711 CHF; Control 812 CHF). *Patient satisfaction:* − No significant differences in patient satisfaction ratings for the final restorations produced in three workflows. *Precision:* − The mean total chairside adjustment time as indicator for clinical precision did not differ significantly for all three groups.
2.	2022, Hashemi et al. [28]	*n* = 10	*n* = 20 [10 + 10]	*n* = 20	Digital: Digital impressions (Trios 3, 3Shape); design (Dental system, 3Shape) and milling (Amann Girrbach) of screw-retained monolithic ZrO2 iFDPs (Katana translucent, Kuraray); clinical evaluation. Conventional: Conventional impressions (Panasil, Kettenbach GmbH & Co.), fabrication of stone models and lab-scan (Atos Core 5 Mp 80 mm; Rev. 02; GOM GmbH); metal casting abutments for full-contour waxing, cut-back and cast with cobalt-chromium alloy; veneering of the framework; clinical reevaluation.	*Accuracy of IOS:* − No significant differences between both workflows for impression accuracy, framework adaptation, and passivity. *Esthetic:* − Subjective assessment by the patients (the mean VAS score was 8.4 ± 0.97 for the conventional technique and 8.6 ± 0.52 for the digital technique, with no significant difference; *p* = 0.684). *Time efficiency:* − No significant difference in the mean clinical time between the two techniques (*p* = 0.444). − Mean laboratory time was shorter for digital workflows (*p* < 0.001). − No differences in the occlusal adjustment time (*p* = 0.143). − Total fabrication time was significantly shorter for the digital workflow (*p* < 0.001).

CAD/CAM, computer-aided design/computer-aided manufacturing; CHF, Swiss Francs; IOS, intraoral optical scanning; LS2, lithium disilicate; PMMA, polymethyl methacrylate; vs., versus; VAS, visual analogue score; ZrO_2_, zirconium dioxide.

## Data Availability

All relevant data are presented in this manuscript. No additional data source is available.

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
