# Peer review of "The Complete Digital Workflow in Fixed Prosthodontics Updated: A Systematic Review"

_healthcare, 2023, doi:10.3390/healthcare11050679_

Round 1

Reviewer 1 Report

Well done and comprehensive article. Good evaluation of the Literature and of the different kind of studies on digital workflow. Discussions can be expanded slightly to improve readers understanding and give them guidelines.

Author Response

Dear Reviewer

Thank you very much for your valuable time and for the positive evaluation of the manuscript. We very much appreciate your effort and your kind words.

We expanded the sections Discussion and Conclusions accordingly.

Reviewer 2 Report

The clinical relevance of the study remains limited by the relatively small number of restorations and the reduced follow up interval and criteria, addressing in particular the elements related to clinical and laboratory time savings and patient comfort in the initial phase. However, these elements have been highlighted by the authors and I believe that the synthesis is scientifically correct

Author Response

Dear Reviewer

Thank you very much for your valuable time and for the positive evaluation of the manuscript. We very much appreciate your effort and your kind words.

Indeed, digital workflows have been trend words in dentistry for a few years now; all the more astonishing that the scientific evidence is still very scanty. This is exactly what we wanted to express.

Reviewer 3 Report

I find the manuscript very well written and presented, and it is clear and provides much clear information about the topic. Digital technology is overcoming the analogic, and it is feasible to think that analogic prosthetic work will be very residual in a few years.

It would be handy to investigate the different production methods and their quality and the prognostic result of the fabricated restorations. Still, of course, that could be the topic of another research.

It could also be interesting to make a meta-analysis of the reported results, trying to find more solid results with larger sample size. I feel actual reported results come from small samples that are probably enough to see trends but unreliable results to implement new digital techniques.

Congratulations to the authors.

Author Response

Dear Reviewer

Thank you very much for your valuable time and for the positive evaluation of the manuscript. We very much appreciate your effort and your kind words.

Indeed, it is also in our interest to shed more light on the darkness. A follow-up review is planned to investigate the different production methods and their quality and the prognostic result of the fabricated restorations. At the moment, however, the number of (qualitative) clinical studies is still small. And YES, we could not agree more to create a meta-analysis.

Reviewer 4 Report

The manuscript is well-written and informative. It clearly states the aim, methods, results, and conclusions of the systematic review. The authors have conducted a rigorous and comprehensive search of the literature and have applied appropriate criteria for selecting and appraising the studies. The review provides valuable insights into the current evidence on digital versus conventional workflows for fixed dental prostheses.

Overall, the review is comprehensive, and well-designed. The authors have followed the PRISMA guidelines and have reported their methods and results transparently. The tables are clear and informative. The only weakness of this review is the lack of included studies after applying the inclusion/exclusion criteria.

Some suggestions to make the minor change before being published:

1.      In p.18, “The proprietary digital workflow 3Shape was shown to be more time efficient than the conventional workflow, while the proprietary digital workflow Dental Wings required more time. The cost analysis was in favor of the digital workflow, with significantly lower production costs for completely digital fabricated FDPs [30]. Based on the same study population, patient-centered outcomes and clinical performance was also investigated.”

It might be nice to mention the names of the actual scanners in each group, rather than just the company, especially those two scanners are both old-generation products. This would help readers to understand better how different technologies affect workflow efficiency and cost-effectiveness.

2.      The limitation of this review, which includes a limited number of studies comparing different digital workflows or different types of FDPs, was not discussed in detail in discussion session. It would be helpful to address this gap in knowledge and suggest directions for future research in this field.

Author Response

Dear Reviewer

Thank you very much for your valuable time and for the positive evaluation of the manuscript. We very much appreciate your effort and your kind words.

Issue-1

In p.18, “The proprietary digital workflow 3Shape was shown to be more time efficient than the conventional workflow, while the proprietary digital workflow Dental Wings required more time. The cost analysis was in favor of the digital workflow, with significantly lower production costs for completely digital fabricated FDPs [30]. Based on the same study population, patient-centered outcomes and clinical performance was also investigated.”

It might be nice to mention the names of the actual scanners in each group, rather than just the company, especially those two scanners are both old-generation products. This would help readers to understand better how different technologies affect workflow efficiency and cost-effectiveness.

>> Thank you for this valuable recommendation. We have added the names of the intraoral scanners accordingly.

Issue-2

The limitation of this review, which includes a limited number of studies comparing different digital workflows or different types of FDPs, was not discussed in detail in discussion session. It would be helpful to address this gap in knowledge and suggest directions for future research in this field.

>> Also thanks for the second valuable recommendation. This helps a lot to increase the quality and comprehensibility for the readers. We have added sections in the Discussion and Conclusions accordingly (please see the modifications in track changes mode).